# Do Empathic Individuals Behave More Prosocially? Neural Correlates for Altruistic Behavior in the Dictator Game and the Dark Side of Empathy

**DOI:** 10.3390/brainsci11070863

**Published:** 2021-06-29

**Authors:** Michael Schaefer, Anja Kühnel, Franziska Rumpel, Matti Gärtner

**Affiliations:** 1Medical School Berlin, 12247 Berlin, Germany; anja.kuehnel@medicalschool-berlin.de (A.K.); matti.gaertner@medicalschool-berlin.de (M.G.); 2Otto-von-Guericke Business School Magdeburg, 39106 Magdeburg, Germany; franziska.rumpel@das-studium.de

**Keywords:** empathy, social, altruistic, dictator game, rTPJ, fMRI

## Abstract

Do empathic individuals behave more prosocially? When we think of highly empathic individuals, we tend to assume that it is likely that those people will also help others. Most theories on empathy reflect this common understanding and claim that the personality trait empathy includes the willingness to help others, but it remains a matter of debate whether empathic individuals really help more. In economics, a prominent demonstration that our behavior is not always based on pure self-interest is the Dictator Game, which measures prosocial decisions in an allocation task. This economic game shows that we are willing to give money to strangers we do not know anything about. The present study aimed to test the relationship between dispositional empathy and prosocial acting by examining the neural underpinnings of prosocial behavior in the Dictator Game. Forty-one participants played different rounds of the Dictator Game while being scanned with functional magnetic resonance imaging (fMRI). Brain activation in the right temporoparietal junction area was associated with prosocial acting (number of prosocial decisions) and associated with empathic concern. Behavioral results demonstrated that empathic concern and personal distress predicted the number of prosocial decisions, but in a negative way. Correlations with the amount of money spent did not show any significant relationships. We discuss the results in terms of group-specific effects of affective empathy. Our results shed further light on the complex behavioral and neural mechanisms driving altruistic choices.

## 1. Introduction

One of the most remarkable abilities of our brain is to understand the thoughts and feelings of our conspecifics (and perhaps to some extent even of some animals). The concept of empathy tries to describe this capability. Although it has gained increased attention in the last years, there is still no clear single definition of what empathy is. Most researchers agree that empathy involves both cognitive as well as affective dimensions, allowing us to understand the thoughts of another as well as vicariously experience his or her feelings (e.g., [1,2]). For example, the widely used questionnaire IRI to measure dispositional empathy describes empathy in affective (empathic concern and personal distress) and cognitive dimensions (perspective taking, fantasy) [3].

Does empathy drive prosocial behavior? Even though many studies addressed this question, it remains a matter of debate (e.g., [4,5]). Perhaps one of the earliest ideas suggesting that empathy may play a key role for prosocial behavior goes back to moral philosopher (and founder of modern economics) Adam Smith, who argued in his “Theory of Moral sentiments” that “Empathy is the source of our fellow-feeling for the misery of others, that is by changing places in fancy with the sufferer, (it is) that we come either to conceive or to be affected by what he feels” [6]. Prosocial behavior can be defined in many ways. Here we define the term as helping others at a cost to the self [7,8].

According to the empathy-altruism hypothesis, empathy motivates prosocial behavior, suggesting that individuals with higher empathy would act more altruistic and care about the welfare of others [9,10,11]. While some researchers found support for this theory and showed, for example, that empathic concern is linked to prosocial behavior [8,11,12], others found less evidence for this relationship. For example, Vachon et al. found only weak relations between empathy and aggression [13]. As the authors stated, this finding is surprising given that “empathy is a core component of many treatments for aggressive offenders”. Furthermore, Jordan et al. found that the concern for others is a predictor for prosocial behavior, whereas empathy does either not or negatively predict altruistic actions (donations to a charitable purpose) [14]. The measure “concern for others” was examined using a new empathy index developed by the authors. It describes empathic concern similar to the scale known from the IRI [3]. The authors argued that empathic concern is psychologically different from “a more narrow sense of empathy defined as feeling what others feel” [14].

Assuming there is a relationship between empathy and altruistic behavior, this does not necessarily have to be linked to empathically feeling the other and being concerned about him or her. There may be other reasons than empathic concern to behave prosocially. For example, empathic individuals may help because of egoistic reasons, for example, avoiding social or self-punishments such as guilt and shame (empathic-specific punishment theory) or reducing one’s own discomfort [15,16]. Furthermore, individuals may behave altruistically because of (social or self) rewards associated with prosocial behavior (honor, pride) (empathic-specific reward hypothesis) [15]. 

However, the relationship between degrees of empathy and social behavior remains unclear only with respect to the average population. It is well known that individuals with low or a lack of empathy (e.g., psychopaths) are characterized by antisocial [17].

Most of the previous studies measuring empathy and altruistic behavior employed an approach in which participants watch others in pain or in a situation needing help (e.g., [18,19]). Neuroimaging studies also focused on paradigms in which the participants could alleviate the distress of concrete individuals they know or that have been described before. For example, FeldmanHall examined the neural correlates for altruism by letting participants witness others in pain and then asking them whether they want to ease their pain. They demonstrated that empathic concern (rather than personal distress) motivates costly altruism [8]. Similarly, it has been shown that when seeing others in pain we tend to help and empathize, especially with group members, a behavior that was accompanied with insula, anterior cingulate cortex (ACC), and medial prefrontal cortex activity [18,20]. Gallo et al. showed a causal role of the somatosensory cortex in prosocial behavior when individuals witness another person’s suffering [21]. However, these paradigms may bear some disadvantages, because the interaction dimension here may stimulate cognitions such as avoiding social punishments or reappraisals as mentioned before. In addition, empathy may be mixed with compassion. 

However, brain imaging studies on decision making have revealed neural mechanisms underlying prosocial behavior, not only when observing individuals needing help but also in more abstract situations (e.g., [22,23,24,25,26]. Several brain areas have been identified that play crucial roles for social behavior. For example, studies addressing theory-of-mind approaches highlighted the role of the medial prefrontal cortex (mPFC), the anterior temporal poles, the right temporoparietal junction area (rTPJ)/posterior superior temporal sulcus (STS) [25,27]. Different roles for these brain areas with respect to mentalizing have been suggested. The mPFC may represent a decoupling mechanism, the rTPJ/posterior STS region is related to detection of agency, and the temporal poles have been linked to social knowledge in the form of scripts [28]. An interesting study demonstrated different roles of the ventral and the dorsal mPFC in social behavior tasks. Whereas the ventral mPFC was engaged in affective processes associated with compassion, the dorsal mPFC represented cognitive operations in this social interaction (such as the selection of the intensity of a retaliation) [29]. 

Furthermore, economics also examined altruistic behavior. For example, in economics it is well known that we often help others even when those individuals are not in a situation where they need help (or where we can ease their pain), sometimes without knowing or seeing them at all. One example is an economic allocation game called the Dictator Game (DG). The DG is one of the most prominent experiments to demonstrate that our behavior is not always based on pure self-interest. The DG is similar to the classic ultimatum game, which addresses cooperation behavior by playing games in which a certain good (usually money) has to be distributed between two players [30,31]. In the DG the task is similar. Again, one player (the dictator) has to distribute a fixed sum of money between himself and the recipient, but here the recipient has no power to punish the proposer or to refuse the money. Therefore, the proposer acts as a dictator. In this way, giving money to the other is not affected by strategic considerations (e.g., the fear of paybacks). Thereby, this paradigm seems to minimize factors touched by the empathic-specific reward or punishment hypotheses. Furthermore, this approach minimizes the demanding impact (e.g., by raising feelings of compassion) of the to-be-helped person. In contrast to the assumption in classical rational choice models in economy, dictators do not always prefer more money to less money. Typical games show that the dictators donate about 20–30% of the money [32]. 

Given that we described prosocial behavior as helping others at a cost to the self (de Waal 2008), we used the DG as a measure of prosocial behavior in order to shed further light on the relationship between empathy and altruism and its neural substrates. We scanned the brain activity of 41 participants while playing different rounds of the DG and tested possible relationships with the personality trait empathy. Based on previous studies, we hypothesized that empathic concern and personal distress would affect altruistic behavior in the DG (e.g., [8]). 

Which brain areas may be related to altruistic behavior in our paradigm? Previous research has shown that costly altruism in the DG has been reported to be linked to an involvement of the rTPJ [33], which is known to play a crucial role for understanding how others feel and think [34,35,36]. Speer and Boksem have demonstrated that prosocial decisions in the DG engage brain areas representing the neural substrate of the theory of mind (TPJ, MPFC and left medial temporal gyrus) [33]. This network seems to facilitate our social understanding. In particular, the rTPJ area has been related to attention and to taking into account other individual’s intentions in moral decisions [35,36]. Thus, we hypothesized that prosocial behavior in our task would engage regions known to be related with prosocial behavior in the DG, in particular the right TPJ area. Furthermore, we control for activation in the insula cortices. Given the results of previous work on a contribution of the mPFC on cognitive operations with respect to behavior in a social task, we also assume that the mPFC might be activated during prosocial decisions in the DG. 

Given that previous studies reported gender effects for trait and state empathy (e.g., suggesting higher empathy scores for females [37], we included sex as an additional variable in our statistical models.

## 2. Materials and Methods 

### 2.1. Participants

The study included 41 right-handed native German participants (26 females, mean age of 21.98 ± 2.76 years). It adhered to the Declaration of Helsinki and was approved by the local human subjects’ committee. All participants gave written informed consent to the study and had no neurological or psychiatric history.

### 2.2. Procedure 

While scanning their brain activity in the MRI, participants were asked to play different rounds of the DG. The DG is similar to the Ultimatum Game (UG), which tests cooperative behavior by the allocation of a certain good (in most studies, a fixed amount of money) between two players (one responder and one provider). Responders who experienced unfair offers in the UG have the opportunity to punish the proposer’s unfair behavior [30,31]. During the DG, the proposer similarly divides money between himself and an unknown responder, but in contrast to the UG, the responder has no chance to punish the proposer for his behavior. In this sense, the proposer has a position that has been described as a dictator: He (or she) is the only one to decide how to split the money and the responder, who he (or she) will never meet, will be passive. In economics, the DG has been used to demonstrate that human behavior is not always motivated by self-interest. 

In the current experiment, we used binary DGs based on previous literature [38,39]. In each round, the participant was provided with an amount of 15 Euros, which he or she had to divide between themselves and the recipient. The participant could choose one of two options of payoffs, one in favor of the recipient and one in favor of the dictator. For example, the dictator had to choose between two options. In the first option, he or she would keep €7.80 for himself and allocate €7.20 to the recipient (selfish option). In the second option, he would keep €7.20 for himself and give €7.80 (prosocial option). Participants used a button box in their right hand to make their decisions. The recipient had no chance to react or comment on the dictator’s decisions. The player were informed that they will never meet their partner. The decision tasks used different distribution options to vary the potential decision conflict. Participants were informed that 30% of the games (in a randomized way) were analyzed and used for paying out the earned money proportionately by crediting test person hours.

The dictator game lasted for 12 s, then there was a break of 12 s before the next trial started. Earlier responding did not start the next trial. The experiment comprised four runs, with a total of 60 dictator games (see Figure 1). Participants were permitted to take short breaks between the runs and were made familiar with the task before starting the experiment. Visual images were back-projected to a screen at the end of the scanner bed close to the subject’s head. Participants viewed the visual stimuli through a mirror mounted on the birdcage of the receiving coil. Foam cushions were placed around the side of the subject’s head to minimize head motion. 

On a separate day, we asked the participants to complete a personality questionnaire. Subjects had to complete a German version of the Interpersonal Reactivity Index (IRI) [3,40]. The 28-item questionnaire is widely known to measure trait empathy (e.g., [41,42]) and describes a cognitive and an affective dimension of empathy. Each dimension has two subscales. The cognitive dimension includes the scale perspective (PT, representing the tendency to think from another perspective) and fantasy (FS, the ability to transpose oneself into the feelings and actions of fictional characters in books, plays, or movies). The affective dimension compromises empathic concern (EC, describing feelings of compassion or sympathy for others) and personal distress (PD, the tendency to experience aversive feelings in response to distress in others). 

### 2.3. FMRI Data Acquisition, Image Preprocessing, and Analysis

MRI scanning was conducted on 3T Siemens Tim Trio scanner (Siemens, Germany). BOLD responses were acquired with echoplanar T2-weighted images (TR = 2 s, TE = 35 ms, flip angle = 80 degrees, FOV = 224 mm, number of slices = 32, voxel size = 3.125 × 3.125 mm, slice thickness = 3.5 mm). High-resolution T1-weighted structural images were recorded using an MP-RAGE sequence prior to the functional runs (TR = 1650 ms, TE = 5 ms). We used Statistical Parametric Mapping Software (SPM12, Wellcome Department of Imaging Neuroscience, University College London, London, UK) to analyze the data. 

Preprocessing steps included motion correction of fMRI images (spatial realignment to the mean image), coregistration, normalization into a standard anatomical space (MNI, Montreal Neurological Institute template, isotropic 3 mm voxels), and smoothing (8 mm FWHM Gaussian kernel). Furthermore, high-pass temporal filtering with a cutoff of 128 s was applied to remove low frequency drifts.

Statistical analysis was done using multiple regressions with the hemodynamic response function modeled in SPM. We first examined data on an individual subject level (fixed-effects model, condition prosocial and egoistic decisions). Time window for the first-level analyses covered the time of presenting the DG (12 s epoch length). The resulting parameter estimates for each regressor at each voxel went into a second-level analysis (random-effects model). 

In this second-level analysis, we first compared brain responses when subjects decided prosocial relative to brain activations when deciding for the selfish option (and vice versa) in order to test our hypothesis about the neural underpinnings of the DG. To further examine whether empathy measures are linked with those brain responses, we then used individual scores of the IRI subscales (EC, PD, FS, and PT) as covariates of interest in a second-level analysis for parametric regressions of this contrast (prosocial relative to selfish decisions).

We report active regions at *p* < 0.05 corrected for multiple comparisons over the whole brain (family-wise (FWE) corrected at cluster level) and activity with small volume correction for a priori regions of interest (ROIs) (at *p* < 0.05, FWE corrected within these ROIs). The small volume correction was applied to ROIs within a sphere of a 15 mm radius. These regions were based on previous research on empathy and altruism and include bilateral anterior insula, the rTPJ, and the mPFC. Coordinates for those ROIs were taken from FeldmanHall et al. [8,29]. Anatomical interpretation of the results was done by using the SPM anatomy toolbox. 

Behavioral data were tested by standard multiple linear regression analyses. All four IRI dimensions and gender went simultaneously into the model to examine whether empathy predicts altruistic behavior. 

## 3. Results 

### 3.1. Behavioral Results 

Table 1 shows mean scores of IRI. IRI dimension EC correlated with FS (r = 0.34, *p* = 0.03; Pearson correlation, two-sided). 

Three participants were excluded prior to further data analysis due to technical reasons during fmri scanning (e.g., loss of behavioral data). Behavioral results demonstrated that participants decided in 50.86% of all allocation games to give away the biggest part of the money. Thus, in about half of the choices they behaved in a prosocial, altruistic way. Behavior in the DG resulted in altruistic decisions with a mean of €4.28 (credited by test person hours). 

Pearson correlations between prosocial acting (the number of social minus selfish decisions) and empathy showed negative significant correlations with empathic concern (r = −0.40, *p* = 0.01, two-sided) and personal distress (r = −0.32, *p* = 0.05, other empathy dimensions: *p* > 0.10). 

To further test the relationship between empathy and prosocial acting, we computed a linear regression analysis, in which all four empathy dimensions (EC, PD, PT, FS) and sex went simultaneously into the model. The results revealed a significant model (R = 0.54, adj. R^2^ = 0.18, F(5,37) = 2.64, *p* = 0.04) and demonstrated that the empathy subscore EC was a significant negative predictor for prosocial behavior of our participants (beta = −0.48, *p* = 0.009). PD revealed a trend for significance (beta = −0.27, *p* = 0.09). Gender also showed a trend (beta = −0.29, *p* = 0.09). Other empathy scales failed to show significant effects (see Table 2 and Figure 2). 

We then tested whether the results are caused by the many repetitions of our study. Therefore, we considered only the first three DG games to measure prosocial behavior. Results confirmed previous findings by showing a negative relationship of empathic concern with altruistic behavior (Pearson correlations; empathic concern: r = −0.18, *p* = n.s., personal distress: r = −0.24, *p* = n.s.). 

Next, we tested whether the difference between money given away and money taken may be related to empathy measures. Results revealed no significant correlations (all *p* > 0.10). 

### 3.2. FMRI Results during Prosocial Acting and Empathy 

Brain responses during prosocial relative to selfish decisions revealed a cluster in rTPJ/posterior STS (FWE, corrected at *p* < 0.05). We did not find brain activity in the anterior insula (even at a lenient threshold of 0.005, uncorrected) (see Figure 3 and Table 3). 

We then used individual scores of the IRI subscale EC as a covariate of interest in a second-level analysis for the contrast prosocial relative to selfish decisions. Results revealed activation in rTPJ/posterior STS (corrected at FWE, *p* < 0.05, see Table 4 and Figure 4). PD linked to prosocial activation revealed no brain activations (uncorrected results at *p* < 0.001 show activity in right precentral gyrus and putamen). PT and FS demonstrated no significant activation (uncorrected results showed ACC activation for PT). 

When examining empathy subscales as covariates of interest for selfish decisions (relative to prosocial acting), we did not find significant activations for EC, PD, and PT. FS showed activation of left postcentral gyrus and rTPJ region (left postcentral gyrus: z = 4.53, 347 voxels; rTPJ: z = 3.45, 6 voxels, *p* < 0.05, FWE corrected).

## 4. Discussion

The present study aimed to investigate whether empathic personality traits affect prosocial behavior and its underlying neural substrates. Results demonstrated engagement of the rTPJ/posterior STS area when participants decided in an altruistic way. Self-reported empathy (empathic concern and personal distress) was linked to this activity but showed a negative relationship with altruistic behavior. 

In the present study, prosocial or altruistic behavior was examined by employing the DG, which is well known in economics to demonstrate that, in contrast to traditional rational choice models, our behavior is not always based on pure self-interest. We found that prosocial decisions in the DG were associated with an activation of the rTPJ/posterior STS. This is in line with numerous other studies showing a role for this brain area in social perception and behavior, in particular, mentalizing, perspective taking, Theory of Mind, and detecting what another person is feeling and thinking [28,34,35,36]. Both low- (discrimination between self and other) as well as high-level sociocognitive operations (emphasize with others) have been related with the rTPJ [43]). Furthermore, the posterior STS has been described as a hub for a distributed brain network for social perception [44,45]. It has been observed that the posterior STS is engaged in the perception of social signals [46]. Moreover, it has been argued that the TPJ-pSTS area might have played a crucial role for the evolution of social abilities in humans [47]. 

In contrast to other studies measuring prosocial behavior [8,18], it seems remarkable that we did not find brain activation in insula. We explain the lack of the engagement of this brain area by the specific task of the DG. In previous studies addressing helping behavior, participants witnessed other individuals, for example, in painful situations, and had the opportunity to help [8,18]. Here participants were asked to give money to someone who they did not know (and would never meet) without a face-to-face situation. Thus, the perspective-taking processes that may underly prosocial behavior in this paradigm may be more abstract and not based on observed emotions or thinking on the emotional state of concrete others.

The specific version of the DG we used may also explain why our results are not in line with the empathy-altruism hypothesis. This theory argues that empathic concern is necessary to feel with others and therefore motivates prosocial behavior [9,10,11]). Our results showed that cognitive empathy was not related to helping behavior, whereas affective empathy (in particular empathic concern) was even negatively linked to altruistic behavior. Again, this may be explained by the missing emotional content of the task, due to not having a face-to-face situation with a concrete person.

Our results confirm previous behavioral studies reporting a lack of relationship between empathy and the prosocial acting in the DG. Lönnquvist and Walkowitz experimentally induced empathy in a DG but did not find an effect of empathy on prosocial behavior in the DG [48]. Similarly, Artinger et al. examined whether individual differences in empathy influence prosocial behavior in the DG. They reported no effects on behavior in the DG or the classic Ultimatum Game [49]. Rose et al. investigated prosocial behavior in aging and found no relationships (or even a negative relationship) with the DG (although a positive relationship was found when positive psychological information about the recipient in the DG was given, but even then, only for young participants) [50]. In contrast to the abovementioned studies, Edele et al. reported positive links of affective empathy with altruistic sharing in the DG, but did not use the affective dimension of the IRI [51]. Moreover, participants in the Edele et al. study played the DG only once and used a different type of DG compared with our study. Furthermore, the DG was framed within a social interaction task, which might have caused the different results compared with our study that asked for decisions on altruistic sharing in a very repetitive way without social frames.

Our results demonstrate the complex nature of helping behavior. Highly empathic individuals in our experiment showed less prosocial behavior in the DG. This is not necessarily a contradiction to the concept of empathy. One could speculate that empathic participants are focused on the well-being and care for others they know or at least can imagine, but that they are not willing to give money to strangers they do not know at all. Similarly, Hein et al. showed that when witnessing someone in pain it is more likely that we are going to help him or her when this individual belongs to our own group [18]). However, it has also been demonstrated that our empathic brain can be trained [52]. Nevertheless, the observation that we often emphasize with members of our group has been used as an argument against the idea to praise and promote empathy. For example, Bloom claimed that there may be many studies showing that empathy can cause prosocial behavior, but empathy can also motivate cruelty and aggression [4]. Thus, there may be dark sides of empathy [53]). 

Although it has been argued that feelings of concern might be raised through cognitive empathy [9,54]), our study (as well as other recent studies, see above) did not find a relationship of cognitive empathy with helping behavior in the DG. Thus, concrete empathic or affective concern for others may be crucial to motivate prosocial behavior. The paradigm we used in the current study provided no possibility to be concerned for the other player; hence, empathy does not seem to play a role for prosocial decisions here. Moreover, we found that empathic concern (and personal distress) had even a negative impact on prosocial behavior. This seems to point to group-specific or selfish processes. If we do not know the others, empathic concern seems to make us keep the money to our group or to ourselves. Thus, when we do not know the others, empathy (empathic concern and personal distress) may even result in less altruistic behavior. 

This is also supported by recent findings of a relationship between neuroticism and the tendency not to behave prosocially [55,56,57]. For example, Guo et al. examined factors affecting online prosocial behavior. They demonstrated that extraversion, agreeableness, and conscientiousness influenced prosocial behavior in a positive way, while neuroticism was a negative predictor. The authors also confirm our results by reporting that empathic concern was not mediating these effects, because online prosocial behavior does not involve face-to-face interaction [57]. 

What may be the practical implications of this study? Considering that previous studies such as Vachon et al. found only weak links between empathy and aggressive behavior, the question arises whether it makes sense to offer empathy training, for example, for aggressive offenders [13]. In light of our data and previous results, it might be more promising to enhance concrete feelings of concern rather than trying to broadly enhance empathy [14]. 

Several limitations of this study have to be addressed. First, the role of social desirability is unclear. However, the present study may be less affected by this factor than other studies. Since the DG was replicated many times, subjects had to decide quickly, and were alone while making their decision. Second, a major limitation is that empathy personality traits were only measured as self-reported empathy. Again, these results may be prone to social desirability factors. Third, in our version of the DG participants were allowed only to decide between two distribution options, they could not allocate the money freely. This may limit the comparison with other studies. Furthermore, the absolute number of prosocial decisions may be only a weak marker of social behavior in contrast to the amount of money participants may give to others. However, when calculating the difference between money taken and money given away in our paradigm, we did not find any effects. Fourth, we used only the DG as a measure of prosocial behavior. It would be very interesting to examine whether altruistic sharing in the DG differs from helping behavior when witnessing, for example, someone in pain. Fifth, our results are based on a sample of students. We do not know whether the results are different when a more representative sample would have been tested. Last, our results are based partly on a ROI approach. This can be a problem with respect to the statistical analyses [58,59,60]. Thus, the definition of ROIs should be well identified a priori and corrected for multiple comparisons. However, results of the present study were corrected and justified based on previous research. 

Taken together, surprisingly the present results do not show that empathy personality traits motivate prosocial acting in the DG. Future studies are needed to further unravel both the complex nature of helping behavior and the personality construct of empathy. 

## Figures and Tables

**Figure 1 brainsci-11-00863-f001:**
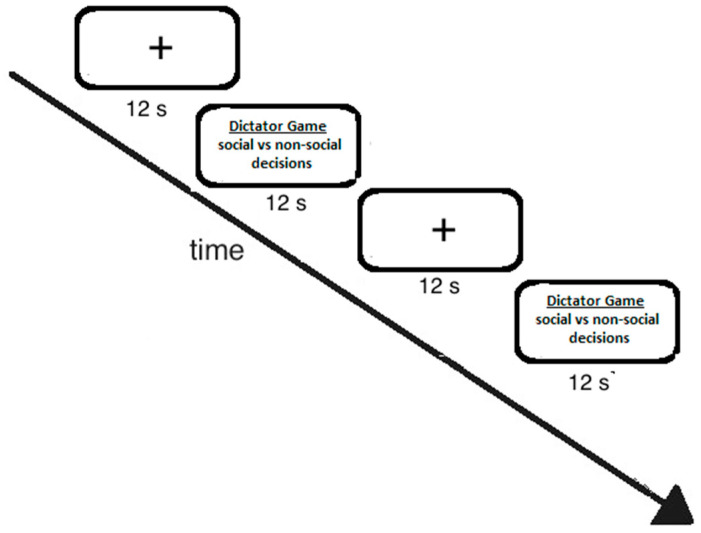
Experimental design. See text for further details.

**Figure 2 brainsci-11-00863-f002:**
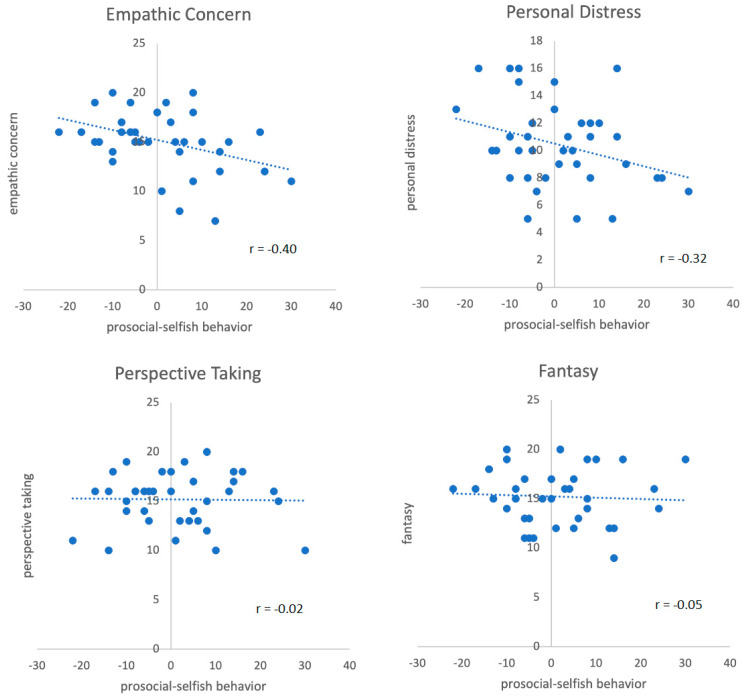
Scatterplots of behavioral responses (prosocial minus selfish decisions) and empathy subscales (Pearson correlations). The figure demonstrates that individuals with high affective empathy (empathic concern and personal distress) tend to be less prosocial in the DG. Cognitive empathy (perspective taking and fantasy) was not related to prosocial behavior.

**Figure 3 brainsci-11-00863-f003:**
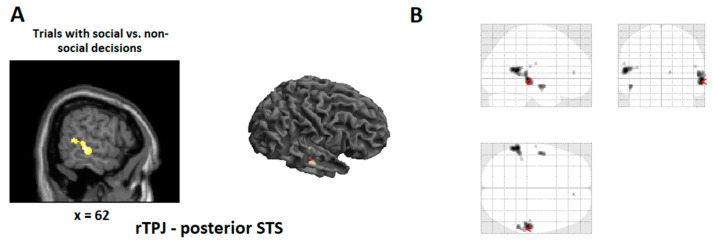
Statistical maps showing brain activation for trials with social relative to non-social decisions. (**A**): Areas of significant fMRI signal change are shown as color overlays on the T1-MNI reference brain (at *p* < 0.005, uncorrected, for picture purpose only). (**B**): Glass brain depicting same contrast as in A (at *p* < 0.005, uncorrected). Even at the very lenient threshold of *p* < 0.005 (uncorrected) we found no anterior insula activation.

**Figure 4 brainsci-11-00863-f004:**
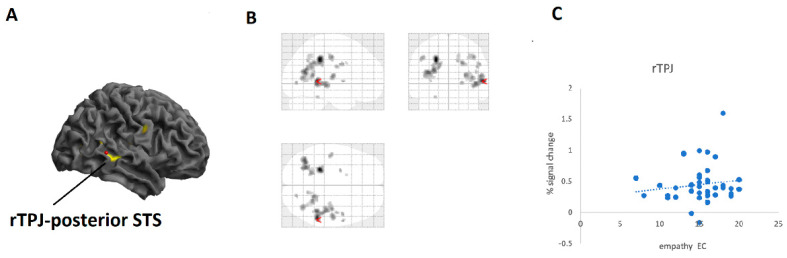
(**A**): Brain responses for trials with social relative to trials with non-social decisions predicted by the empathy subscale empathic concern. Results revealed brain activation in rTPJ-posterior STS (at *p* < 0.001, uncorrected, for picture display only). (**B**): Glass brain depicting same contrast as in A (at *p* < 0.001, uncorrected). (**C**): Scatterplot showing empathic concern correlated with peak activation in rTPJ-posterior STS brain region (Pearson’s correlation: r = 0.14).

**Table 1 brainsci-11-00863-t001:** Results of personality questionnaires IRI.

		Mean ± Standard Deviation
Empathy Personality Questionnaire IRI	Empathic Concern	15.32 ± 2.99
	Personal Distress	10.49 ± 3.04
	Perspective Taking	15.29 ± 2.62
	Fantasy	15.44 ± 2.86

**Table 2 brainsci-11-00863-t002:** Regression analyses of prosocial acting with empathy subscales as predictors. Significant values in bold.

Model					Coefficients (Standardized)		
R	R^2^	adj. R^2^	ANOVA		Betas	T	Sign.
0.54	0.29	0.18	F (5,37) = 2.64, *p* = 0.04	EC:	−0.48	2.77	*p* = **0.009**
				PD:	−0.27	1.73	*p* = 0.09
				PT:	0.09	−0.59	*p* = 0.56
				FS:	0.05	−0.32	*p* = 0.75
				sex:	−0.29	1.77	*p* = 0.09

EC = empathic concern; PD = personal distress; PT = perspective taking; FS = fantasy.

**Table 3 brainsci-11-00863-t003:** Results of the contrast of brain responses during prosocial relative to selfish acting (*p* < 0.05, FWE corrected).

	Brain Region	Peak MNI Location (x, y, z)	Peak z-Value	Number of Voxels
prosocial behavior vs. selfish behavior	rTPJ/posterior STS	62 −30 −2	3.35	128
selfish vs. prosocial behavior	-	-	-	-

**Table 4 brainsci-11-00863-t004:** Results of individual scores of the IRI subscales as covariates of interest for parametric regressions of the contrast prosocial relative to selfish acting (*p* < 0.05, FWE corrected).

Covariates	Brain Region	Peak MNI Location (x, y, z)	Peak z-Value	Number of Voxels
EC	r. TPJ/posterior STS	56 −42 4	4.44	511
PD	-			
PT	-			
FS	-			

EC = empathic concern; PD = personal distress; PT = perspective taking; FS = fantasy.

## Data Availability

The data presented in this study are available on request from the corresponding author.

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
