# Peer review of "Do Empathic Individuals Behave More Prosocially? Neural Correlates for Altruistic Behavior in the Dictator Game and the Dark Side of Empathy"

_brainsci, 2021, doi:10.3390/brainsci11070863_

Round 1

Reviewer 1 Report

In this study, individuals are solving the dictator game in the MR scanner, and are later tested on their individual empathy scores using a questionnaire. The authors find that activity in the right TPJ is higher when participants decide in a prosocial compared to antisocial way, and that empathy scores correlate negatively with the number of trials individuals chose a prosocial compared to antisocial action. This paper touches an interesting topic, but it needs methodological and conceptual revision.

Major concerns

  • There has to be some clarification of terminology. The authors write towards the end of the introduction that they investigate "altruistic empathic behavior" whereas before they state that they want to test the relation between empathy and prosocial (altruistic) behavior. Terminology is therefore mixed here. Also, prosocial behavior is defined very late in the intro. In addition, the authors cite an article that found a relation between empathy and concern for others and prosocial behavior but not between empathy and prosocial behavior, without clearly defining these terms. I would therefore suggest to first clarify terminology early on in the introduction before discussing the relationship between the different measures in detail. This can also be used to clarify why the dictator game was chosen as test here.
  • Given the task, I am not sure that the total number of prosocial decisions - the total number of antisocial decisions is the correct variable of interest. To my understanding, in each trial, there was a certain amount of money that could be allocated to the participant or the other person. However, the amount of difference differed between trials. This may influence future decisions. That is, if, for example, the participant decided to give away 5 Euros and to take 5.10 Euros, he/she will be motivated to also take more money in the upcoming trial. If, however, the participant just took 10 Euros and only gave away 2 Euros, he/she may feel that next time he/she should act prosocial. I therefore think that the critical variable here is the DIFFERENCE between the MONEY TAKEN and the MONEY GIVEN AWAY (in Euros) rather than number of trials acted prosocial versus antisocial. Please conduct the same analyses and correlations with this variable (i.e., correlation to empathy, predictor in fMRI model).
  • The authors need to provide a cognitive model on how they think their definition of empathy and prosoial behavior relates to brain activations. At the moment, there is just one sentence in the introduction with no further reference to why these areas should be activated. Based on present literature, it should be lined out which areas take which role in the process, and which brain areas would be expected to be involved in the present task.

Minor concerns

Introduction

  • The authors need to mention that a lack of empathy links to more antisocial behavior in the case of psychopaths; the unclear relationship between degrees of empathy and social behavior therefore only concerns the average population but not those individuals with very low empathy.
  • The statement that neuroscience studies mostly investigate altruistic behavior by studying the observation of pain is a wrong oversimplification; neuroscience studies on decision making regularly investigate the neuronal basis of decision processes such as social or altruistic behavior that does not involve observation of others or the observation of pain; this literature should be mentioned and cited by the authors
  • There is only one sentence about the expected brain areas to be involved in the process, without any explanation of how these brain areas actually relate to prosocial behavior or empathy. It would be important to explain the cognitive model behind these brain activations.

Methods/Results

  • It would be good to know how large the ROIS were for the small volume corrections. Could the authors please provide the number of voxels within each mask.
  • It is not clear to me why the big five inventory was correlated to the brain data. There is no mentioning of this inventory or any hypotheses associated with this in the introduction. I would suggest deleting this section.
  • The figures need to be reworked: they should not have such bold headings such as "prosocial versus selfish" due to the concerns explained in the major concerns. Rather, the figures should precisely reflect what was calculated, i.e., trials with social behavior versus trials with non-social behavior. 

Discussion

  • The discussion is at parts a bit unscientific. I.e., speculating of a lacking "feeling of warm glow" in this context is pure speculation; individuals were not asked how they felt during the task. Please remove these and similar speculations.
  • In the discussion, suddenly the concept of generosity is introduced, which is not mentioned before. This has to be properly introduced if the authors want to use it for their arguments. What is the definition of generosity and where is the difference to prosocial behavior?

Author Response

Comments of reviewer #1:

Thanks for the positive and helpful comments on our manuscript.

  1. „ There has to be some clarification of terminology. The authors write towards the end of the introduction that they investigate "altruistic empathic behavior" whereas before they state that they want to test the relation between empathy and prosocial (altruistic) behavior. Terminology is therefore mixed here. Also, prosocial behavior is defined very late in the intro. In addition, the authors cite an article that found a relation between empathy and concern for others and prosocial behavior but not between empathy and prosocial behavior, without clearly defining these terms. I would therefore suggest to first clarify terminology early on in the introduction before discussing the relationship between the different measures in detail. This can also be used to clarify why the dictator game was chosen as test here.."

We now revised the introduction to first clearly define the terminology of the different measures.

We added and changed in the introduction page 7:

“Which brain areas may be related to altruistic behavior in our paradigm?”

Page 3:

“Prosocial behavior can be defined in many ways. Here we define the term as helping others at a cost to the self (de Waal, 2008).”

Page 7:

“Given that we described prosocial behavior as helping others at a cost to the self (de Waal, 2008), we used the DG as a measure of prosocial behavior…”

We also refer to the paper by Jordan et al. more in detail and describe more clearly how they defined empathy concern and prosocial behavior (page 4):

“Furthermore, Jordan et al. found that the concern for others is a predictor for prosocial behavior, whereas empathy does either not or negatively predict altruistic actions (donations to a charitable purpose) (Jordan, Amir, & Bloom, 2016). The measure “concern for others” was examined using a new empathy index developed by the authors. It describes empathic concern similar to the scale known from the IRI (Davis, 1983). The authors argue that empathic concern is psychologically different from empathy in a narrow sense.”

  1. „Given the task, I am not sure that the total number of prosocial decisions - the total number of antisocial decisions is the correct variable of interest. To my understanding, in each trial, there was a certain amount of money that could be allocated to the participant or the other person. However, the amount of difference differed between trials. This may influence future decisions. That is, if, for example, the participant decided to give away 5 Euros and to take 5.10 Euros, he/she will be motivated to also take more money in the upcoming trial. If, however, the participant just took 10 Euros and only gave away 2 Euros, he/she may feel that next time he/she should act prosocial. I therefore think that the critical variable here is the DIFFERENCE between the MONEY TAKEN and the MONEY GIVEN AWAY (in Euros) rather than number of trials acted prosocial versus antisocial. Please conduct the same analyses and correlations with this variable (i.e., correlation to empathy, predictor in fMRI model).”

This is an interesting point. Social behavior may be less expressed in the absolute number of social decisions, but in the amount of money we give away. We now considered the difference between money taken and money given away in our analyses and reanalyzed our data with respect to this variable. Our results showed no correlations with empathy measures or fMRI data (correlation with EC r = 0.03, PD r = -0.04, PT r = -0.07, FS r = 0.17). This lack of results may be explained by the paradigm we used in the current study. In contrast to other DGs participants in our version of the DG had only the choice between two fixed options of distribution (ego or altruistic), they could not freely decide how much money they would give away.

 We added on page 13:

“Next, we tested whether the difference between money given away and money taken may be related to empathy measures. Results revealed no significant correlations (all p > 0.10). “

In addition, we added in the discussion section on page 19:

“Furthermore, the absolute number of prosocial decisions may be only a weak marker of social behavior in contrast to the amount of money participants may give to others. However, when calculating the difference between money taken and money given away in our paradigm, we did not find any effects.”

  1. „The authors need to provide a cognitive model on how they think their definition of empathy and prosoial behavior relates to brain activations. At the moment, there is just one sentence in the introduction with no further reference to why these areas should be activated. Based on present literature, it should be lined out which areas take which role in the process, and which brain areas would be expected to be involved in the present task.”

Thanks for drawing our attention to this point. We agree with the reviewer that this issue should be explained more in detail. We added on pages 5 and 6:

“Which brain areas may be related to altruistic behavior in our paradigm? Previous research reported a network of brain regions including the dorsal ACC and bilateral anterior insula when witnessing someone in pain (e.g., (Singer et al., 2004)). However, in the DG participants do not watch someone in pain. Furthermore, due to the same reason, neural mechanisms underlying the caregiving model (e.g., ACC, amygdala, periaqueductal gray (Preston et al., 2013)), should not be central to the paradigm used in this study. In contrast, costly altruism in the DG has been reported to be linked to an involvement of the right temporoparietal junction area (rTPJ) (Speer & Boksem, 2019), which is known to play a crucial role for understanding how others feel and think (Koster-Hale, Saxe, Dungan, & Young, 2013; Young, Camprodon, Hauser, Pascual-Leone, & Saxe, 2010; Young, Dodell-Feder, & Saxe, 2010). Speer and Boksem have demonstrated that prosocial decisions in the DG engage brain areas representing the neural substrate of the theory-of-mind (bilateral TPJ, medial prefrontal cortex (MPFC) and left medial temporal gyrus)  (Speer & Boksem, 2019). This network seems to facilitate our social understanding. In particular, the rTPJ area has been related to attention and to considering other individual’s intentions in moral decisions (Young, Camprodon, et al., 2010; Young, Dodell-Feder, et al., 2010). Thus, we hypothesized that prosocial behavior in our task would engage regions known to be related with prosocial behavior in the DG, in particular the right TPJ area. Furthermore, we control for activation in the insula cortices and the ACC.”

Minor:

- The authors need to mention that a lack of empathy links to more antisocial behavior in the case of psychopaths; the unclear relationship between degrees of empathy and social behavior therefore only concerns the average population but not those individuals with very low empathy.

We added on page 4:

“However, the relationship between degrees of empathy and social behavior remains unclear only with respect to the average population. It is well known that individuals with low or a lack of empathy (e.g., psychopaths) are characterized by antisocial behavior (Viding, McCrory, & Seara-Cardoso, 2014).”

-„The statement that neuroscience studies mostly investigate altruistic behavior by studying the observation of pain is a wrong oversimplification; neuroscience studies on decision making regularly investigate the neuronal basis of decision processes such as social or altruistic behavior that does not involve observation of others or the observation of pain; this literature should be mentioned and cited by the authors.”

We agree with the reviewer that this point should be revised. We added on page 5:

“However, brain imaging studies on decision making have revealed neural mechanisms underlying prosocial behavior not only when observing individuals needing help but also in more abstract situations (e.g.,  (Burnett, Sebastian, Cohen Kadosh, & Blakemore, 2011; Lockwood, Apps, & Chang, 2020; Spitzer, Fischbacher, Herrnberger, Grön, & Fehr, 2007; Stanley & Adolphs, 2013; Steinbeis, Bernhardt, & Singer, 2012)). Several brain areas have been identified that play crucial roles for social behavior. For example, studies addressing theory-of-mind approaches highlighted the role of the medial prefrontal cortex (mPFC), the anterior temporal poles, the right temporoparietal junction area (rTPJ) / posterior superior temporal sulcus (STS) (Burnett et al., 2011; Mahy, Moses, & Pfeifer, 2014). Different roles for these brain areas with respect to mentalizing has been suggested. The mPFC may represent a decoupling mechanism, the rTPJ / posterior STS region is related to detection of agency, and the temporal poles have been linked to social knowledge in the form of scripts (Frith & Frith, 2003). An interesting study demonstrated different roles of the ventral and the dorsal mPFC in social behavior tasks. Whereas the ventral mPFC was engaged in affective processes associated with compassion, the dorsal mPFC represented cognitive operations in this social interaction (such as the selection of the intensity of a retaliation) (Lotze, Veit, Anders, & Birbaumer, 2007).”

- There is only one sentence about the expected brain areas to be involved in the process, without any explanation of how these brain areas actually relate to prosocial behavior or empathy. It would be important to explain the cognitive model behind these brain activations..

We now revised this paragraph. See above, our response to point 3.

-“Methods/Results: It would be good to know how large the ROIS were for the small volume corrections. Could the authors please provide the number of voxels within each mask.”

We now provided this information (page 11):

The small volume correction was applied to ROIs within a sphere of 15 mm radius.

-It is not clear to me why the big five inventory was correlated to the brain data. There is no mentioning of this inventory or any hypotheses associated with this in the introduction. I would suggest deleting this section.

We followed the suggestion of the reviewer and removed the Big Five data.

-The figures need to be reworked: they should not have such bold headings such as "prosocial versus selfish" due to the concerns explained in the major concerns. Rather, the figures should precisely reflect what was calculated, i.e., trials with social behavior versus trials with non-social behavior. 

We now revised the figures using more precise descriptions.

-Discussion: The discussion is at parts a bit unscientific. I.e., speculating of a lacking "feeling of warm glow" in this context is pure speculation; individuals were not asked how they felt during the task. Please remove these and similar speculations.

We followed the suggestion of the reviewer and removed speculations in the discussion (see now revised discussion section).

-In the discussion, suddenly the concept of generosity is introduced, which is not mentioned before. This has to be properly introduced if the authors want to use it for their arguments. What is the definition of generosity and where is the difference to prosocial behavior?

We removed the term generosity to avoid misunderstandings.

Reviewer 2 Report

This is an important manuscript regarding the neural correlates of empathic and altruistic behavior.

Here are some suggestions to help improve the manuscript.

1) Methods: was any denoising performed to remove physiological noise?

2) The data and figures shown in Figures 1, 2, and 3 are just excellent. However, this reviewer suggests that authors add an additional figure (scatter plot) from the results of a correlation between the individual scores for prosocial behavior (and/or empathic concern) and fMRI statistical map score from a region of interest in the rTPJ. So in other words generate a similar scatter plot as that shown in figure 1 except for behavior versus fMRI signal.

Author Response

Comments of reviewer #2:

We would like to thank the reviewer for the positive and helpful comments on our manuscript.

  1. „Methods: was any denoising performed to remove physiological noise??"

No further preprocessing steps other than described to remove physiological noise was done.

  1. “The data and figures shown in Figures 1, 2, and 3 are just excellent. However, this reviewer suggests that authors add an additional figure (scatter plot) from the results of a correlation between the individual scores for prosocial behavior (and/or empathic concern) and fMRI statistical map score from a region of interest in the rTPJ. So in other words generate a similar scatter plot as that shown in figure 1 except for behavior versus fMRI signal.”

Thanks for point. The revised manuscript now includes a new Figure showing the correlation between brain activation in rTPJ and behavior (see new Fig. 3).

Reviewer 3 Report

brainsci-1249735: Do empathic individuals behave more prosocial? Neural correlates for altruistic behavior in the dictator game and the dark side of empathy

General comments:

The authors here address a highly interesting topic in a highly innovative experiment using functional MRI during a social interaction game. They found right temporopatrietal junction activation associated with prosocial behaviour in the dictator game.

More specific comments:

Introduction: well formulated; very nice to read! The references provided for social interaction fMRI-paradigms are somehow meagre. In addition many important social relevant areas are not described as usually done – such as the STS, the ventral prefrontal cortex, the anterior temporal pole ….

With respect to empathy and pain observation and application there might be an earlier study of interest which emphasized the role of the medial prefrontal cortex for empathic behavior in a joked Taylor paradigm (Lotze et al., 2007: Evidence for a different role of the ventral and dorsal medial prefrontal cortex for social reactive aggression: An interactive fMRI study).

That might also have an impact on the hypotheses: “…that prosocial behavior would engage regions known to be related with empathy and prosocial behavior, in particular the right temporoparietal junction area, the insula cortices, and the ACC.“

Methods: Participants and Testing: sound description; Please describe the actual payment of each participant in reality (for the 30% selected) and for the other 70% participants. fMRI-data acquisition: please indicate slice thickness (32 slices). Please indicate onsets and conditions fort he first level analyses. Please indicate more explicitly the stat. testing performed for the second level analyses. Please explain more precisely how the hypotheses have been tested. A methods Figure would be great to illustrate onsets modelled as conditions of interest in first level analysis. Please indicate whether you use cluster or height corrected FWE whole brain P<0.05 thresholds. See the comment on hypothesis driven tests restricted on ROIs. These should be corrected for multiple comparisons within ROIs but not presented as uncorrected thresholds.

Results: It is sometimes difficult to follow the huge amount of abbreviations. Could you help the reader with a overview of abbreviations? Please also provide footnotes for abbreviations used in the Tables (especially for EC, PD, PT, FS).

Figure 1: ok

Figure 2: rather odd illustration. The authors might describe the location of the effect not only as rTPJ but also as posterior STS.

Figure 3: how can it bee that a Figure thresholded with FWE 0.05 shows a insula activation with a z-value of about 3? It looks that this Figure is thresholded with p<0.001,uncorrected. Again a quite odd Figure in a visualisation quality of the middle 9tees of the last millennium.

Over all Table 4 shows a lot of uncorrected results but the authors had prior hypotheses on ROI-testing. Why didn't they follow that line here?

Discussion: The authors might think about adding some literature and changing the labels of the effects discussed (e.g., TPJ-posterior STS). Limitations: It might be of interest to discuss possible problems by ROI-selection and statistical testing applied here. Especially in the light of the quite restricted selection of hypothesized ROIs and the odd presentation of the results in the Figures. In addition, a glass brain shown with uncorrected thresholding (p<0.001) might be a good way to allow a review for all effects present in the statistical testing.

Minor Issues:

Page 3, line 111: close second brackets:

(e.g., (FeldmanHall et al. 2015)

Author Response

Comments of reviewer #3:

We would like to thank the reviewer for the positive and helpful comments on our manuscript.

  1. „ The references provided for social interaction fMRI-paradigms are somehow meagre. In addition many important social relevant areas are not described as usually done – such as the STS, the ventral prefrontal cortex, the anterior temporal pole …..” 

In the revised version we now considered more references for social interaction fMRI-paradigms. Furthermore, we now also address other brain areas discussed in those papers (STS; PFC, anterior temporal pole). We added on pages 5 and 6:

“However, brain imaging studies on decision making have revealed neural mechanisms underlying prosocial behavior not only when observing individuals needing help but also in more abstract situations (e.g.,  (Burnett et al., 2011; Lockwood et al., 2020; Spitzer et al., 2007; Stanley & Adolphs, 2013; Steinbeis et al., 2012)). Several brain areas have been identified that play crucial roles for social behavior. For example, studies addressing theory-of-mind approaches highlighted the role of the medial prefrontal cortex (mPFC), the anterior temporal poles, the right temporoparietal junction area (rTPJ) / posterior superior temporal sulcus (STS) (Burnett et al., 2011; Mahy et al., 2014). Different roles for these brain areas with respect to mentalizing has been suggested. The mPFC may represent a decoupling mechanism, the rTPJ / posterior STS region is related to detection of agency, and the temporal poles have been linked to social knowledge in the form of scripts (Frith & Frith, 2003).”

  1. „ With respect to empathy and pain observation and application there might be an earlier study of interest which emphasized the role of the medial prefrontal cortex for empathic behavior in a joked Taylor paradigm (Lotze et al., 2007: Evidence for a different role of the ventral and dorsal medial prefrontal cortex for social reactive aggression: An interactive fMRI study).

That might also have an impact on the hypotheses: “…that prosocial behavior would engage regions known to be related with empathy and prosocial behavior, in particular the right temporoparietal junction area, the insula cortices, and the AC

Thanks for this drawing our attention to this interesting paper. We now address these findings on page 5:

“An interesting study demonstrated different roles of the ventral and the dorsal mPFC in social behavior tasks. Whereas the ventral mPFC was engaged in affective processes associated with compassion, the dorsal mPFC represented cognitive operations in this social interaction (such as the selection of the intensity of a retaliation) (Lotze et al., 2007).”

And on page 7:

“Given the results of previous work on a contribution of the mPFC on cognitive operations with respect to behavior in a social task, we also assume that the mPFC might be activated during prosocial decisions in the DG.”

  1. „ Methods: Participants and Testing: sound description; Please describe the actual payment of each participant in reality (for the 30% selected) and for the other 70% participants. fMRI-data acquisition: please indicate slice thickness (32 slices). Please indicate onsets and conditions fort he first level analyses. Please indicate more explicitly the stat. testing performed for the second level analyses. Please explain more precisely how the hypotheses have been tested. A methods Figure would be great to illustrate onsets modelled as conditions of interest in first level analysis. Please indicate whether you use cluster or height corrected FWE whole brain P<0.05 thresholds. See the comment on hypothesis driven tests restricted on ROIs. These should be corrected for multiple comparisons within ROIs but not presented as uncorrected thresholds.

The revised manuscript now added information on actual payment, data acquisition, and stats.

Page 12:

“Behavior in the DG resulted in altruistic decisions with a mean of 4.28 € (credited by test person hours).”

Page 10:

“BOLD responses were acquired with echoplanar T2-weighted images (TR = 2 sec, TE = 35 ms, flip angle = 80 degrees, FOV = 224 mm, number of slices = 32, voxel size = 3.125 x 3.125 mm, slice thickness = 3.5 mm).”

On pages 10 and 11:

“We first examined data on an individual subject level (fixed-effects-model, condition prosocial and egoistic decisions). Time window for the first level analyses covered the time of presenting the DG (12 seconds epoch length).”

On page 11:

“In this second-level analysis we first compared brain responses when subjects decided prosocial relative to brain activations when deciding for the selfish option (and vice versa) in order to test our hypothesis about the neural underpinnings of the DG. To further examine whether empathy measures are linked with those brain responses, we then used individual scores of the IRI subscales (EC, PD, FS, and PT) as covariates of interest in a second level analysis for parametric regressions of this contrast (prosocial relative to selfish decisions).”

In addition, we added a figure to help understanding our design (new Fig. 1).

On page 11 (see also our new limitation with respect to ROIs below):

“We report active regions at p < 0.05 corrected for multiple comparisons over the whole brain (family-wise (FWE) corrected at cluster level) and activity with small volume correction for a priori regions of interest (ROIs) (at p < 0.05, FWE corrected within these ROIs).”

  1. „ Results: It is sometimes difficult to follow the huge amount of abbreviations. Could you help the reader with a overview of abbreviations? Please also provide footnotes for abbreviations used in the Tables (especially for EC, PD, PT, FS).”

We followed the suggestion of the reviewer and included an overview of abbreviations and also provide footnotes for tables.

  1. Figures: Figure 2: rather odd illustration. The authors might describe the location of the effect not only as rTPJ but also as posterior STS.

We now improved the Figures and also refer to a more correct description of rTPJ / posterior STS in the figures and in the main text.

  1. Figure 3: how can it bee that a Figure thresholded with FWE 0.05 shows a insula activation with a z-value of about 3? It looks that this Figure is thresholded with p<0.001,uncorrected. Again a quite odd Figure in a visualisation quality of the middle 9tees of the last millennium.

We regret the confusion about the figures. The revised version now includes improved figures depicting more clearly the thresholds we used for displaying the activations here. Furthermore, we removed uncorrected data from the tables. However, in order to inform about activation clusters with a more lenient threshold we now included glass brain activations, as suggested by the reviewer.

  1. Over all Table 4 shows a lot of uncorrected results but the authors had prior hypotheses on ROI-testing. Why didn't they follow that line here?

We agree with the referee to remove the uncorrected results.

  1. Discussion: The authors might think about adding some literature and changing the labels of the effects discussed (e.g., TPJ-posterior STS).

We followed the suggestion of the reviewer and added some more literature (with respect to TPJ-posterior STS) in the discussion section. In addition, we now refer to this activation as TPJ – posterior STS (see also our new tables).

We added on page 15:

“Furthermore, the posterior STS has been described as a hub for a distributed brain network for social perception (Dasgupta, Tyler, Wicks, Srinivasan, & Grossman, 2017; Lahnakoski et al., 2012). It has been observed that the posterior STS is engaged in the perception of social signals (Isik, Koldewyn, Beeler, & Kanwisher, 2017).  Moreover, it has been argued that the TPJ-pSTS area might have played a crucial role for the evolution of social abilities in humans (Patel, Sestieri, & Corbetta, 2019).”

  1. Limitations: It might be of interest to discuss possible problems by ROI-selection and statistical testing applied here. Especially in the light of the quite restricted selection of hypothesized ROIs and the odd presentation of the results in the Figures. In addition, a glass brain shown with uncorrected thresholding (p<0.001) might be a good way to allow a review for all effects present in the statistical testing.

We now added a further limitation with respect to hypothesis-drive tests restricted on ROIs. Furthermore, we added a glass brain showing uncorrected results in Figs. 3 and 4.

We added on page 19:

“Last, our results are based partly on a ROI approach. This can be a problem with respect to the statistical analyses (Lyon, 2017; Poldrack & Mumford, 2009; Vul, Harris, Winkielman, & Pashler, 2009). Thus, the definition of ROIs should be well identified a priori and corrected for multiple comparisons. However, results of the present study were corrected and justified based on previous research.”

Minor Issues:

Page 3, line 111: close second brackets: (e.g., (FeldmanHall et al. 2015)

 Done.

Round 2

Reviewer 1 Report

I thank the authors for revising their manuscript; the revision improved the quality of the manuscript significantly. A few concerns remain:

  1. The authors do not show a relationship between spent money and brain activation, the correlation only holds for the number of times where participants acted prosaically. This should be included in the abstract.
  2. When defining empathy, the authors write the sentence "The authors argue that empathic concern is psychologically different from in a narrow sense" -> one word seems missing, it is also not clear what "in a narrow sense" means here
  3. When describing the brain areas that may be involved in the task, the authors write that prior studies found that the ACC is involved in seeing pain in others, but that this, however, is not relevant to the present study because here, people do not see others in pain. These sentences are therefore superfluous. I would suggest removing them. Similar to the next sentence.

Author Response

Response to reviews

Comments of reviewer #1:

We would like to thank the reviewer for the important and helpful comments on our revised manuscript.

  1. „ The authors do not show a relationship between spent money and brain activation, the correlation only holds for the number of times where participants acted prosaically. This should be included in the abstract..."

We agree with the reviewer that this specification of our results is important and should be noted in the abstract. We added in the abstract:

“Brain activation in right temporoparietal junction area was associated with prosocial acting (number of prosocial decisions) and associated with empathic concern. Behavioral results demonstrated that empathic concern and personal distress predicted the number of prosocial decisions, but in a negative way. Correlations with the amount of money spent did not show any significant relationships “

  1. „When defining empathy, the authors write the sentence "The authors argue that empathic concern is psychologically different from in a narrow sense" -> one word seems missing, it is also not clear what "in a narrow sense" means here”

We now corrected and improved this statement to:

“The authors argue that empathic concern is psychologically different from “a more narrow sense of empathy defined as feeling what others feel” (Jordan, Amir, & Bloom, 2016).”

  1. „When describing the brain areas that may be involved in the task, the authors write that prior studies found that the ACC is involved in seeing pain in others, but that this, however, is not relevant to the present study because here, people do not see others in pain. These sentences are therefore superfluous. I would suggest removing them. Similar to the next sentence.”

Thanks for this note. We removed these statements (page 7, last paragraph).